# Structure of protein O-mannose kinase reveals a unique active site architecture

Qinyu Zhu[1,2], David Venzke[3,4,5], Ameya S Walimbe[3,4,5], Mary E Anderson[3,4,5], Qiuyu Fu[6], Lisa N Kinch[7], Wei Wang[2,8,9,10], Xing Chen[2,8,9,10], Nick V Grishin[7], Niu Huang[6], Liping Yu[11], Jack E Dixon[12,13,14], Kevin P Campbell[3,4,5]*, Junyu Xiao[1,2]*

[1]The State Key Laboratory of Protein and Plant Gene Research, School of Life Sciences, Peking University, Beijing, China; [2]Academy for Advanced Interdisciplinary Studies, Peking-Tsinghua Center for Life Sciences, Peking University, Beijing, China; [3]Department of Molecular Physiology and Biophysics, Howard Hughes Medical Institute, University of Iowa Roy J and Lucille A Carver College of Medicine, Iowa City, United States; [4]Department of Neurology, University of Iowa Roy J and Lucille A Carver College of Medicine, Iowa, United States; [5]Department of Internal Medicine, University of Iowa Roy J and Lucille A Carver College of Medicine, Iowa, United States; [6]National Institute of Biological Sciences, Beijing, China; [7]Department of Biophysics, University of Texas Southwestern Medical Center, Dallas, United States; [8]Beijing National Laboratory for Molecular Sciences, College of Chemistry and Molecular Engineering, Peking University, Beijing, China; [9]Synthetic and Functional Biomolecules Center, Peking University, Beijing, China; [10]Key Laboratory of Bioorganic Chemistry and Molecular Engineering of Ministry of Education, Peking University, Beijing, China; [11]Medical Nuclear Magnetic Resonance Facility, University of Iowa Roy J and Lucille A Carver College of Medicine, Iowa, United States; [12]Department of Pharmacology, University of California, San Diego, La Jolla, United States; [13]Department of Cellular and Molecular Medicine, University of California, San Diego, La Jolla, United States; [14]Department of Chemistry and Biochemistry, University of California, San Diego, La Jolla, United States

*For correspondence: kevin-campbell@uiowa.edu (KPC); junyuxiao@pku.edu.cn (JX)

**Abstract** The 'pseudokinase' SgK196 is a protein O-mannose kinase (POMK) that catalyzes an essential phosphorylation step during biosynthesis of the laminin-binding glycan on α-dystroglycan. However, the catalytic mechanism underlying this activity remains elusive. Here we present the crystal structure of *Danio rerio* POMK in complex with $Mg^{2+}$ ions, ADP, aluminum fluoride, and the GalNAc-β3-GlcNAc-β4-Man trisaccharide substrate, thereby providing a snapshot of the catalytic transition state of this unusual kinase. The active site of POMK is established by residues located in non-canonical positions and is stabilized by a disulfide bridge. GalNAc-β3-GlcNAc-β4-Man is recognized by a surface groove, and the GalNAc-β3-GlcNAc moiety mediates the majority of interactions with POMK. Expression of various POMK mutants in *POMK* knockout cells further validated the functional requirements of critical residues. Our results provide important insights into the ability of POMK to function specifically as a glycan kinase, and highlight the structural diversity of the human kinome.

## Introduction

The human kinome contains more than 500 eukaryotic protein kinases (EPKs), which regulate a diverse array of cellular processes (*Manning et al., 2002*). Recently, the collection of human kinases has been further expanded by the discovery of novel kinases that function specifically in the secretory pathway (*Dudkiewicz et al., 2013*; *Sreelatha et al., 2015*; *Tagliabracci et al., 2013a*, *2013b*). These proteins are so divergent from the canonical EPKs that they eluded earlier identification and were not included on the kinome tree. Among them, four-jointed, a Golgi kinase, phosphorylates atypical cadherins Fat and Dachsous to regulate planar cell polarity in *Drosophila* (*Ishikawa et al., 2008*). Fam20C is the long-sought physiological casein kinase that phosphorylates many secreted proteins (*Tagliabracci et al., 2012*, *2015*). Fam20B is a xylose kinase that regulates the biosynthesis of proteoglycans (*Koike et al., 2009*; *Wen et al., 2014*). Fam20A lacks intrinsic kinase activity and functions as a positive regulator of Fam20C (*Cui et al., 2015*). The crystal structure of *Caenorhabditis elegans* Fam20C orthologue reveals an atypical kinase architecture remotely related to the EPKs (*Xiao et al., 2013*). Importantly, mutations in the Fam20 proteins cause several diseases in humans including biomineralization defects, underscoring the physiological significance of phosphorylation-regulated processes in the secretory pathway (*Sreelatha et al., 2015*).

Interestingly, two members of the human kinome that were positioned near the root of the kinome tree, have been found to function in the secretory pathway. Vertebrate lonesome kinase (VLK/PKDCC/SgK493) phosphorylates a wide range of extracellular and endoplasmic reticulum (ER) resident proteins on tyrosine residues (*Bordoli et al., 2014*; *Kinoshita et al., 2009*). On the other hand, protein O-mannose kinase (POMK, previously referred to as SgK196) is a carbohydrate kinase like Fam20B, and plays a critical role for the biosynthesis of functional α-dystroglycan (α-DG) (*Yoshida-Moriguchi et al., 2013*).

α-DG is a subunit of the dystroglycan complex, and binds to basement membrane molecules such as laminin to connect the extracellular matrix with the actin cytoskeleton (*Barresi and Campbell, 2006*; *Yoshida-Moriguchi and Campbell, 2015*). α-DG is also a receptor for human pathogens including members of Old World arenaviruses such as the Lassa fever virus (*Cao et al., 1998*; *Jae et al., 2013*). All these functions critically depend on proper glycosylation of α-DG, an elaborate process that involves at least 17 different enzymes (*Yoshida-Moriguchi and Campbell, 2015*). The modification starts with a unique trisaccharide molecule GalNAc-β3-GlcNAc-β4-Man attached to Ser/Thr residues on α-DG (*Yoshida-Moriguchi et al., 2013*). The mannose is then phosphorylated by POMK at the C6 hydroxyl position, to allow further glycan elongation by enzymes including Fukutin, FKRP, TMEM5, B4GAT1, and LARGE (*Figure 1—figure supplement 1*) (*Gerin et al., 2016*; *Inamori et al., 2012*; *Jae et al., 2013*; *Kanagawa et al., 2016*; *Praissman et al., 2016*; *Willer et al., 2014*; *Yoshida-Moriguchi et al., 2010*). *POMK* mutations cause a spectrum of congenital and limb-girdle muscular dystrophies, including the most severe presentation known as the Walker-Warburg syndrome, which is associated with brain and eye abnormalities and death in early childhood (*Di Costanzo et al., 2014*; *Jae et al., 2013*; *von Renesse et al., 2014*).

Despite compelling biochemical evidence in support of the kinase activity of POMK, its catalytic mechanism was puzzling, since it was long considered a pseudokinase. The canonical EPKs feature several highly conserved structural elements that are essential for catalysis, which are exemplified by the cAMP-dependent protein kinase (PKA), including Lys72[PKA] from strand β3 that coordinates ATP, Glu91[PKA] from helix αC that pairs with Lys72[PKA], Asp166[PKA] in the H/YRD motif that serves as the catalytic base, and Asp184[PKA] in the DFG motif that functions as the primary metal-binding residue (*Hanks and Hunter, 1995*; *Jura et al., 2011*; *Taylor and Kornev, 2011*). None of these important residues or motifs were found in POMK (*Eyers and Murphy, 2013*; *Manning et al., 2002*). In this study, we have used a multidisciplinary approach to determine the architecture of the active site and the mechanisms underlying the catalytic and substrate-recognition activities of this unusual kinase. We show the crystal structure of *Danio rerio* POMK in complex with $Mg^{2+}$ ions, ADP, aluminum fluoride, and the GalNAc-β3-GlcNAc-β4-Man trisaccharide substrate. This structure provides a snapshot of the catalytic transition state of this glycan kinase, revealing an unprecedented kinase active site that is established by residues located in non-canonical positions and is stabilized by a disulfide bridge. The structure further reveals that GalNAc-β3-GlcNAc-β4-Man is recognized by a surface groove, and that the GalNAc-β3-GlcNAc moiety mediates the majority of interactions with POMK. We further show by Nuclear Magnetic Resonance (NMR) analysis that GalNAc-β3-GlcNAc-β4-Man

binds to POMK with a dissociation constant of 30.2 μM. Finally, we express various *Homo sapiens* POMK mutants in a *POMK* knockout cell line to validate the functional requirements of critical residues in POMK and to understand disease-causing mutations. Our results consolidate the catalytic function of POMK during the post-translational processing of α-DG, and facilitate a better understanding of the dystroglycanopathies and various physiological systems that depend on dystroglycan.

## Results

### Overall structure of POMK

*Homo sapiens* POMK (HsPOMK) contains a type II transmembrane (TM) domain and a lumenal kinase domain (*Figure 1A*). There are four potential N-linked glycosylation sites in its kinase domain, Asn67, Asn165, Asn220, and Asn235. POMK is highly conserved throughout evolution (*Figure 1—figure supplement 2*). Interestingly, no glycosylation site is present in DrPOMK, which is otherwise 58% identical to HsPOMK in the kinase domain and can efficiently phosphorylate the GalNAc-β3-GlcNAc-β4-Man trisaccharide (*Figure 1—figure supplement 3*). Loss of DrPOMK in *Danio Rerio* also leads to disrupted muscle function (*Di Costanzo et al., 2014*). We crystallized DrPOMK kinase domain and determined its structure in complex with Mg/ADP, aluminum fluoride, and GalNAc-β3-GlcNAc-β4-Man at 2.0 Å resolution (*Table 1*). Seven conserved Cys are present in POMK homologues, and six of them are involved in forming three pairs of disulfide bridges in DrPOMK. Cys53-Cys66 is located in a long loop in the backside of the N-lobe (*Figure 1B*). Cys72-Cys139 connects helix αB and strand β4. Cys201-Cys241 links the catalytic loop with the activation segment. Cys310 alone exists as a free cysteine and is buried in the C-lobe, not exposed to the solvent.

DrPOMK kinase has a bilobal architecture characteristic of EPKs, and can be superimposed onto PKA with a rmsd (root-mean-square difference) of 3.1 Å over 222 aligned Cα atoms. The N-lobe of DrPOMK highly resembles that of PKA, containing a five-stranded β-sheet (β1-β5) coupled to the αC helix (*Figure 2A*, *Figure 2—figure supplement 1*). The C-lobe is more divergent. Although strands β7-β8 and helices αE-αF closely correspond to the equivalent structure elements in PKA, helices αD, αH, and αI of DrPOMK exhibit significant differences in conformation and length. Helix αG is absent in DrPOMK, whereas an α-helix is uniquely present in its catalytic loop (αCL).

### Active site structure

In POMK, a Ser (Ser106$^{DrPOMK}$) occupies the position of the critical Lys72$^{PKA}$ in strand β3 (*Figure 2B*). The critical role of this Lys in DrPOMK is instead served by Lys91$^{DrPOMK}$ located at the beginning of strand β2 that reaches into the active site and interacts with the phosphate groups of ADP (*Figure 3*). Mutation of the corresponding Lys in HsPOMK to Gly completely abolished kinase activity (K93G, *Figure 4A*, *Figure 4—figure supplement 1*). The catalytic activity is restored in a double mutant, K93G/S108K, which reinstalls the Lys in strand β3. In fact, this mutant, having both the Gly-rich loop and the β3 Lys restored to 'normal', has enhanced kinase activity in vitro compared to the wild-type enzyme. Another important Lys involved in nucleotide-binding is Lys208$^{DrPOMK}$ located in helix αCL. Similar to Lys168$^{PKA}$, it interacts with the AlF$_3$ group that mimics the transition state γ-phosphate of ATP (*Figure 3*). Mutation of the equivalent Lys in HsPOMK to Ala also significantly impairs catalysis (K210A, *Figure 4A*).

In most kinases, Glu91$^{PKA}$ forms an ion pair with Lys72$^{PKA}$, which is important for an active kinase. In POMK, Glu91$^{PKA}$ in the αC helix is replaced by a Gly (Gly120$^{DrPOMK}$, *Figure 2B*). Lys91$^{DrPOMK}$ forms an ion pair with Asp227$^{DrPOMK}$, which occupies the position of the DFG Gly (*Figure 3*). However, mutation of the homologous Asp to Gly in HsPOMK has no negative effect on catalysis (D229G, *Figure 4A*). Furthermore, restoration of the DFG Gly and the αC Glu (D229G/G122E, *Figure 4A*) eliminated kinase activity. Thus, unlike in canonical kinases, an ion pair appears not to be required for the activity of POMK.

The catalytic loop and the activation segment, separated by a pair of anti-parallel β-strands (β7-β8), are located in the linker region between helices αE and αF. Compared to PKA, the linkers between helices αE and αF are much longer in POMK, and the catalytic Asp (Asp202$^{DrPOMK}$) is located in a MCD motif in POMK (*Figure 2B*). Mutation of this Asp in HsPOMK to Ala eradicated kinase activity (D204A, *Figure 4A*), corroborating its critical catalytic function. Like the His/Tyr in the

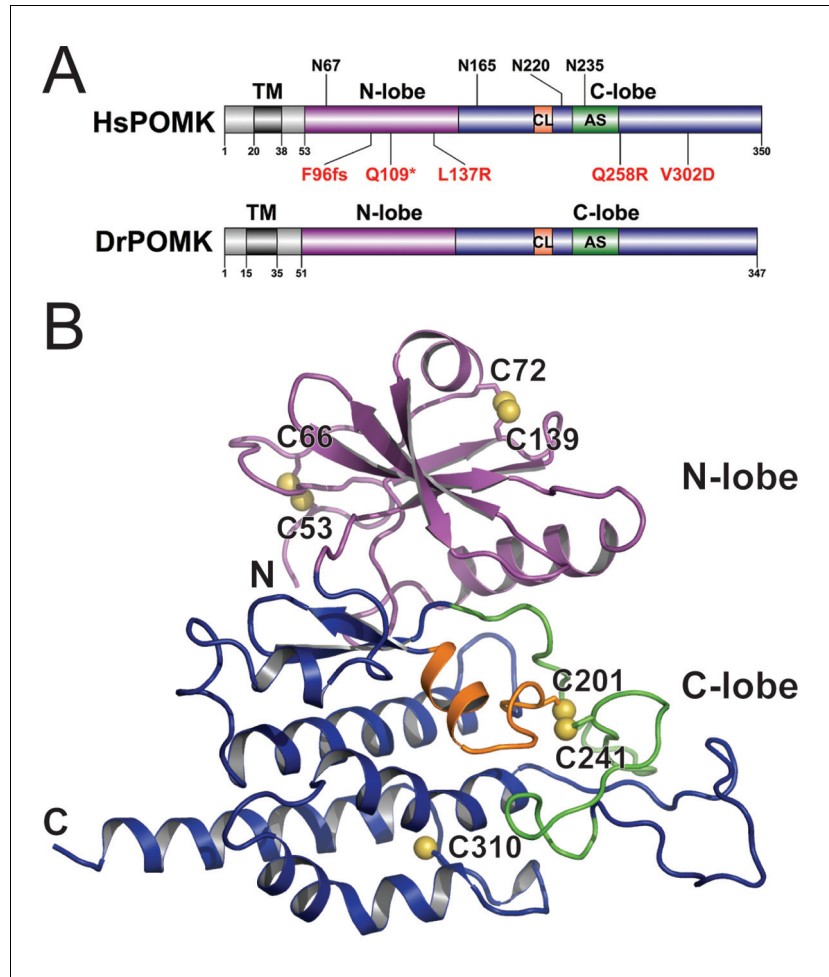

**Figure 1.** Crystal structure of DrPOMK. (**A**) Schematic representation of HsPOMK and DrPOMK depicting the type II TM domain (black), the kinase domain (N-lobe: magenta; C-lobe: blue), the catalytic loop (CL, orange), and the activation segment (AS, green). Predicted N-linked glycosylation sites and mutations found in patients are shown at the top and bottom of HsPOMK, respectively. F96fs is a frame-shift mutation and Q109* is a nonsense mutation. (**B**) Ribbon representation of DrPOMK structure, shown in the same color scheme as *Figure 1A*. The sulfur atoms in the Cys residues are depicted in yellow spheres. The N- and C-termini of the protein are indicated.

The following figure supplements are available for figure 1:

**Figure supplement 1.** A schematic model of the functional α-DG structure and enzymes involved in the glycan biosynthesis.

**Figure supplement 2.** Sequence alignment of POMK kinase domain.

**Figure supplement 3.** POMK homologues phosphorylates GGM-MU.

H/YRD motif, the Met in the MCD is involved in forming the regulatory spine structure that stabilizes the active kinase conformation (*Figure 2A*) (*Taylor and Kornev, 2011*). In kinases that undergo phosphorylation-dependent activation such as PKA, the H/YRD Arg (Arg165[PKA]) interacts with a phosphorylated residue in the activation segment (Thr197[PKA]) to organize the active site (*Figure 3*). In POMK, the MCD Cys (Cys201[DrPOMK]) forms a disulfide bridge with the Cys in the activation segment (Cys241[DrPOMK]). Mutation of these two Cys in HsPOMK to Ala abolished kinase activity (C203A/C244A, *Figure 4A*). This pair of disulfide bridge is also involved in interacting with the trisaccharide substrate as described below.

**Table 1.** Data collection and refinement statistics.

| | Se-Met DrPOMK | DrPOMK in complex with Mg/ADP, AlF₃, and GGM-MU (PDB ID: 5GZA) |
|---|---|---|
| Data collection | | |
| Space group | C2 | P3₂ |
| Cell dimensions | a = 217.95 Å, b = 107.03 Å, c = 151.99 Å, β = 134.2° | a = 70.55 Å, b = 70.55 Å, c = 66.94 Å, |
| Wavelength (Å) | 0.979 | 0.979 |
| Resolution (Å) | 2.85 | 2.0 |
| $R_{merge}$ | 7.8 (56.8) | 11.7 (51.4) |
| $I / \sigma I$ | 17.9 (2.0) | 24.4 (3.2) |
| Completeness (%) | 100 (100) | 100 (100) |
| Multiplicity | 3.8 (3.8) | 7.9 (7.9) |
| Wilson B-factor | 61.4 | 43.1 |
| Refinement | | |
| Unique reflections | | 25147 |
| $R_{work}$ / $R_{free}$ | | 19.7 / 21.8 |
| No. of atoms | | |
| Protein | | 2308 |
| Ligand/ion | | 85 |
| Protein residues | | 298 |
| B-factors | | |
| Protein | | 52.3 |
| Ligand/ion | | 50.3 |
| R.m.s deviations | | |
| Bond lengths (Å) | | 0.009 |
| Bond angles (°) | | 1.137 |
| Ramachandran | | |
| Favored (%) | | 92.4 |
| Allowed (%) | | 7.6 |
| Outliers (%) | | 0 |

Each dataset was collected from a single crystal. Values in parentheses are for highest-resolution shell.

The DFG Asp (Asp184[PKA]) is the primary metal-chelating residue in protein kinases. A DLD motif is present in POMKs, and the first Asp (Asp225[DrPOMK]) adopts a similar position as Asp184[PKA] (*Figure 2B*, *Figure 3*). Mutation of this Asp in HsPOMK to Ala abolished kinase activity (D227A, *Figure 4A*). Like the DFG Phe, the DLD Leu is also part of the regulatory spine (*Figure 2A*). The catalytic loop of POMK is five residues longer than PKA, and contains a short helix αCL. Gln212[DrPOMK] at the C-terminal end of αCL occupies the position of Asn171[PKA], and functions as the second metal-binding residue (*Figure 3*). Mutation of the corresponding Gln in HsPOMK to Ala reduced kinase activity in vitro (Q214A, *Figure 4A*).

## POMK specifically recognizes GalNAc-β3-GlcNAc-β4-mannose

To determine the binding affinity of POMK for the trisaccharide substrate GalNAc-β3-GlcNAc-β4-Man, we synthesized GalNAc-β3-GlcNAc-β4-Man attached to a 4-methylumbelliferyl group (GalNAc-β3-GlcNAc-β4-Man-α-MU or GGM-MU, *Figure 5A*) using chemical and enzymatic methods. We then analyzed the GGM-MU by NMR to confirm the glycan structure, and measured GGM-MU's affinity

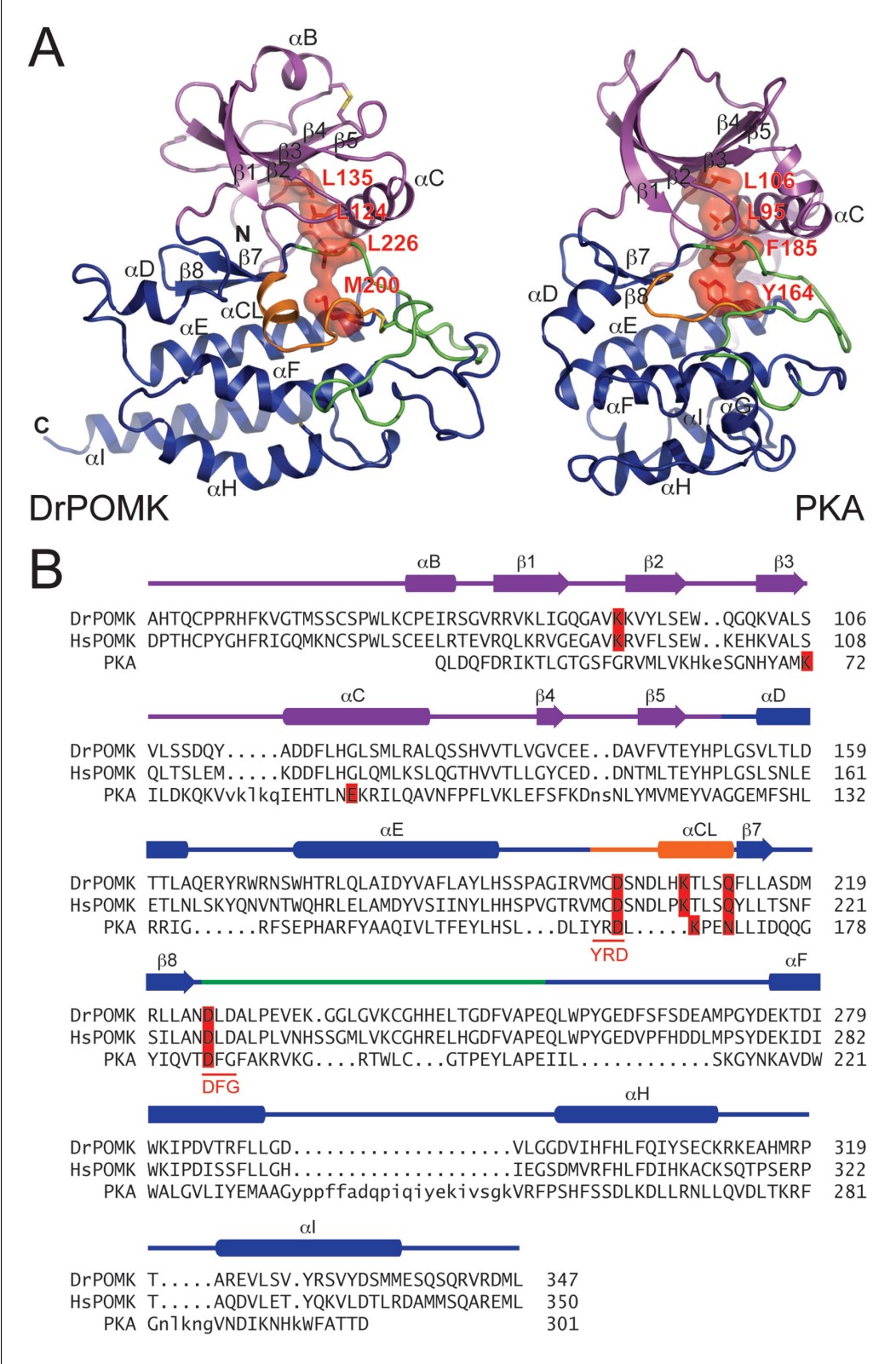

**Figure 2.** Structure comparison between DrPOMK and PKA. (**A**) DrPOMK and PKA structures are shown in the same color scheme as *Figure 1*. Secondary structures of DrPOMK are labeled following PKA convention. The regulatory spines in the two proteins are depicted in red. (**B**) Structure-based sequence alignment of DrPOMK, HsPOMK, and PKA. Residues essential for kinase activity are highlighted in red. The YRD and DFG motifs in PKA are underlined.

*Figure 2 continued on next page*

*Figure 2 continued*

The following figure supplement is available for figure 2:

**Figure supplement 1.** Superposition of DrPOMK and PKA.

to POMK in a manner similar to what we recently reported (*Briggs et al., 2016*). The mannose anomeric proton (Man H1) is well resolved and its intensities were found to decrease with increasing POMK protein concentration (*Figure 5B*). By fitting the intensity changes of the Man H1 peak as a function of POMK concentration, we obtained a dissociation constant of 30.2 µM (*Figure 5C*).

We observed that the peak intensity of MU H3 proton decreased only slightly even when the Man H1 peak is nearly saturated (fully bound) by adding 64.2 µM POMK, indicating that the MU group is mobile and does not interact with the protein strongly (*Figure 5—figure supplement 1*). Therefore, the MU group contribution to the glycan binding affinity is likely small or negligible. This was also observed for the glycan binding protein laminin-α2 LG4-5 when bound to MU-tagged oligosaccharides (*Briggs et al., 2016*).

The trisaccharide is nestled in a groove next to the nucleotide-binding pocket (*Figure 6A*, *Figure 6—figure supplement 1*). The GalNAc-β3-GlcNAc moiety buried 448 Å² (223 Å² from GalNAc; 225 Å² from GlcNAc) solvent-accessible surfaces and accounted for the majority of interactions with DrPOMK. The disaccharide arches over the Cys201-Cys241 disulfide bridge, and is sandwiched by residues including Gly242 and His243 from the front side of the groove, and Asp116 and Ala228 from the back side (*Figure 6B*). Tyr113 shelters the disaccharide from the top. In particular, Ala228 is located at the center on the back side of the groove, and has its side chain pointing to the Gal-NAc-β3-GlcNAc to mediate hydrophobic/Van der Waals interactions. Mutation of the corresponding Ala to Glu in HsPOMK eliminated kinase activity (A230E, *Figure 4A*). Asp116, Cys201, Asn204, Gly242, and His243 form five hydrogen bonds with the GalNAc-β3-GlcNAc, two of which are mediated by main chain groups of Cys201 and Gly242 (*Figure 6B*). Mutation of the Asp116^DrPOMK-

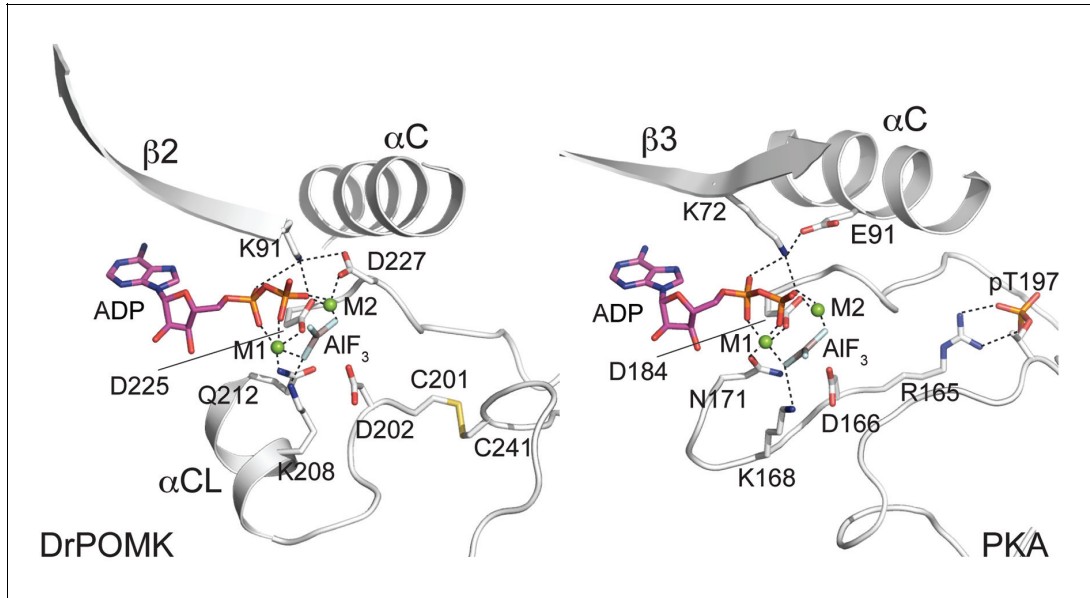

**Figure 3.** Structure of a transition state mimic of DrPOMK reveals residues required for catalysis. Left: an enlarged image of the nucleotide-binding pocket of DrPOMK showing the molecular interactions important for kinase activity. The carbon, nitrogen, oxygen, and sulfur atoms of DrPOMK protein are shown in white, blue, red, and orange respectively. The carbons of ADP are colored in magenta. AlF₃ is shown in sticks. The two Mg²⁺ ions (M1 and M2) are shown as green spheres. Salt bridge and hydrogen bond interactions are shown as dashed lines. Right: the active site of PKA is shown in the same orientation and coloring scheme for comparison (PDB ID: 1L3R).

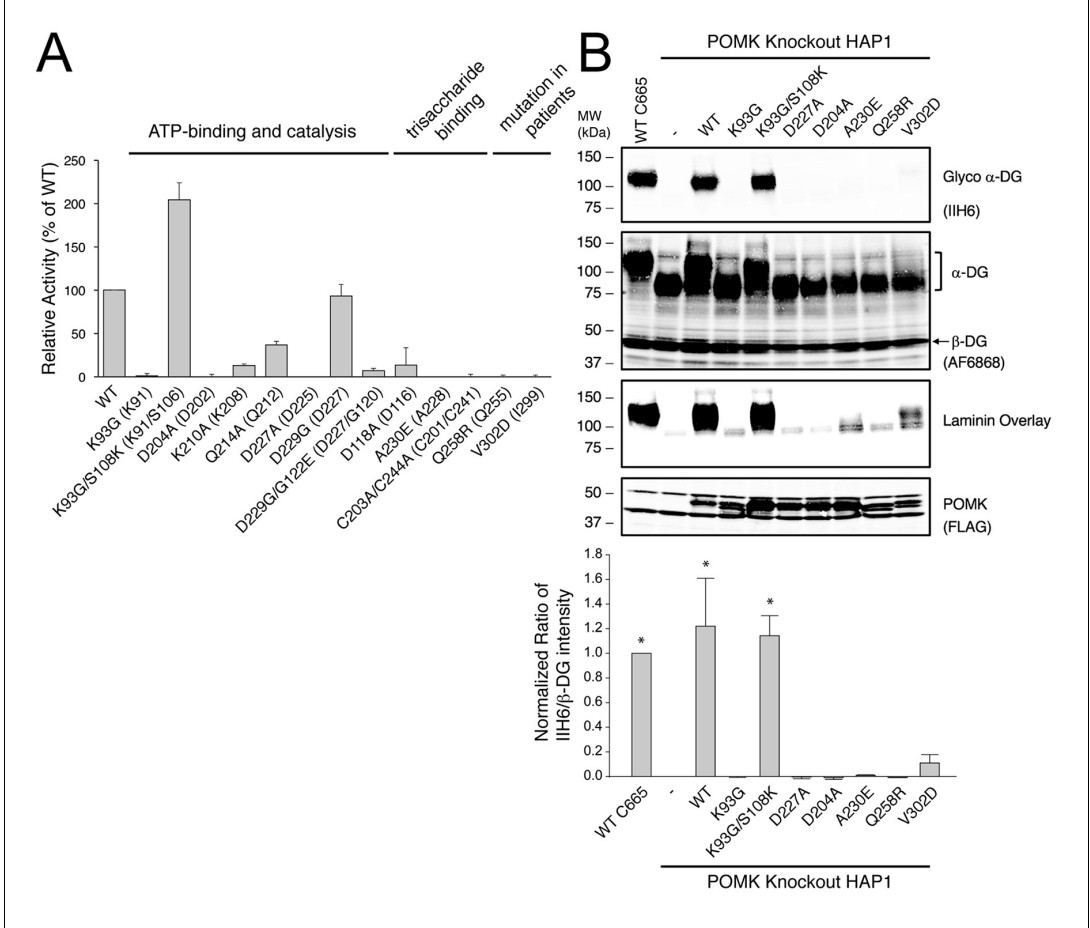

**Figure 4.** POMK mutants have reduced or abolished activity. (**A**) POMK mutants display reduced or abolished kinase activity in vitro. HsPOMK proteins were assayed as described in Materials and methods. Activity of each mutant relative to that of the wild-type enzyme are depicted graphically. Error bars represent standard deviation of three independent experiments. The amino acids in brackets indicate the corresponding residues in DrPOMK. (**B**) POMK mutants are functionally defective in vivo. DG from wild-type C665 (a diploid cell line containing duplicated chromosomes of HAP1) and *POMK* KO HAP1 cells infected with indicated adenoviruses were analyzed by immunoblotting using anti-α-DG-glycan antibody (IIH6) and anti-DG-core antibody (AF6868). The laminin-binding ability of α-DG from these cells was examined using a laminin overlay assay. Expression of POMK was monitored using an anti-Flag antibody. The relative glycosylation level of α-DG was represented by the ratio of IIH6 immunoblot intensity to that of β-DG, normalized to the ratio from wild-type C665 cells (three replicates, error bars representing standard errors of the mean). Asterisks indicate p<0.001 compared to *POMK* KO alone.

The following figure supplements are available for figure 4:

**Figure supplement 1.** Purification of HsPOMK mutants.

**Figure supplement 2.** Un-cropped images of western blotting results shown in *Figure 4B*.

equivalent Asp in HsPOMK to Ala also greatly impaired catalysis (D118A, *Figure 4A*). The residues mentioned above are highly conserved (*Figure 1—figure supplement 2*), suggesting an unchanged substrate preference of POMK during evolution.

The most prominent interaction between the mannose residue and DrPOMK is seen between the Man-O6 hydroxyl group (the phosphoacceptor) and a carboxylate oxygen of Asp202[DrPOMK], the catalytic Asp (*Figure 6B*). This spatial arrangement makes Asp202[DrPOMK] an ideal catalytic base to facilitate phosphoryl transfer from ATP, consistent with the well-documented reaction mechanism of protein kinases (*Adams, 2001*; *Gerlits et al., 2015*). The planar AlF$_3$ group is sandwiched between

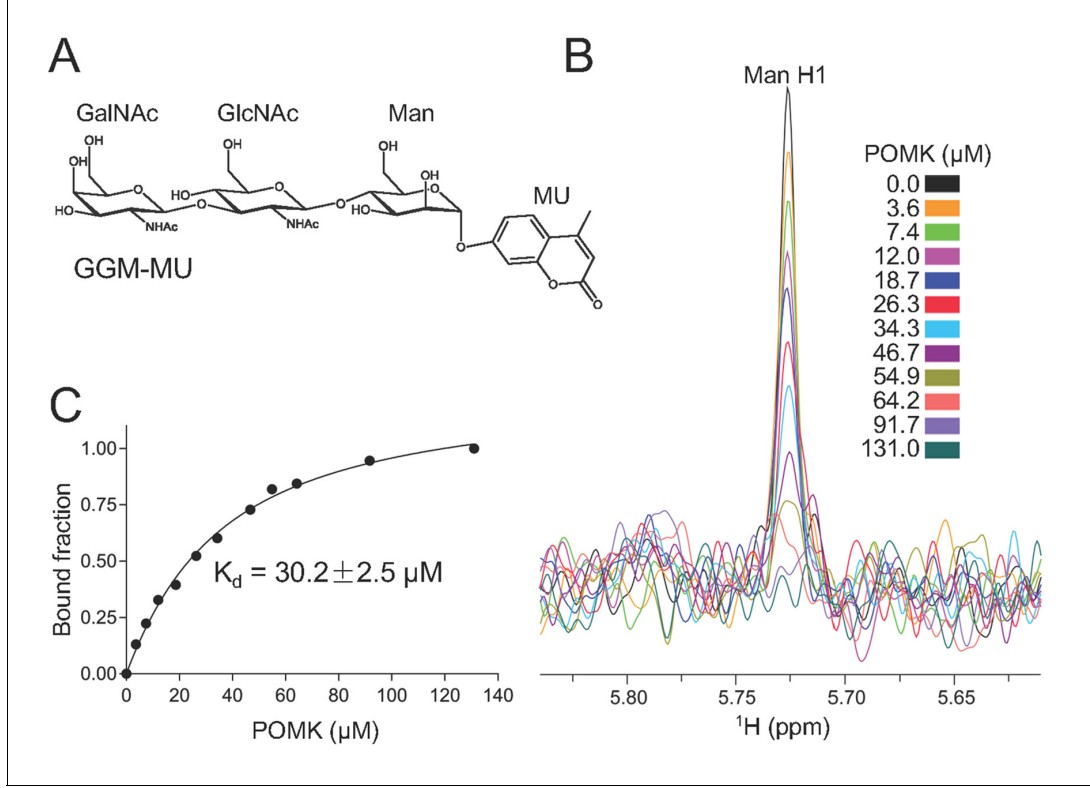

**Figure 5.** NMR analysis of trisaccharide GGM-MU binding to DrPOMK. (**A**) The chemical structure of GGM-MU. (**B**) 1D [1]H NMR spectra of the anomeric region of 10 μM GGM-MU acquired in a Tris buffer containing indicated concentrations of DrPOMK. The anomeric peak derived from mannose (Man H1) is labeled. (**C**) Determination of dissociation constant from the intensity changes of the anomeric peak of Man H1. The standard deviation from data fitting is shown.
The following figure supplement is available for figure 5:

**Figure supplement 1.** 1D [1]H NMR spectra of trisaccharide GGM-MU.

the β-phosphate of ADP and the Man, and is 1.9 Å from both the ADP donor oxygen and Man-O6, mimicking the catalytic transition state.

## POMK mutants are functionally defective in vivo

To further validate the functional requirement of critical residues in POMK, we obtained *POMK* knockout (KO) haploid human HAP1 cells generated using the CRISPR/Cas9 genome-editing technique, in which we expressed various HsPOMK mutants using recombinant adenoviruses in order to evaluate their activity. In the control C665 cells (a diploid cell line containing duplicated chromosomes of HAP1), both the glycoepitope of α-DG (recognized by antibody IIH6) and the α-DG core protein (recognized by antibody AF6868) were detected at ~120 kDa (*Figure 4B*, *Figure 4—figure supplement 2*). The glycan modification on α-DG was severely reduced in the *POMK* KO cells, as suggested by the disappearance of the glycoepitope and the mobility change of the α-DG core protein. Consistently, these cells were defective in binding to laminin in a laminin overlay assay (*Figure 4B*). Expression of wild-type POMK or the K93G/S108K mutant by adenoviruses fully rescued the functional glycosylation of α-DG. In contrast, expression of K93G, D204A, D227A, or A230E did not restore the glycoepitope, the molecular weight, or the laminin-binding ability of α-DG (*Figure 4B*). These results demonstrate the importance of these residues for POMK function in vivo, corroborating our structural observations and in vitro biochemical analyses.

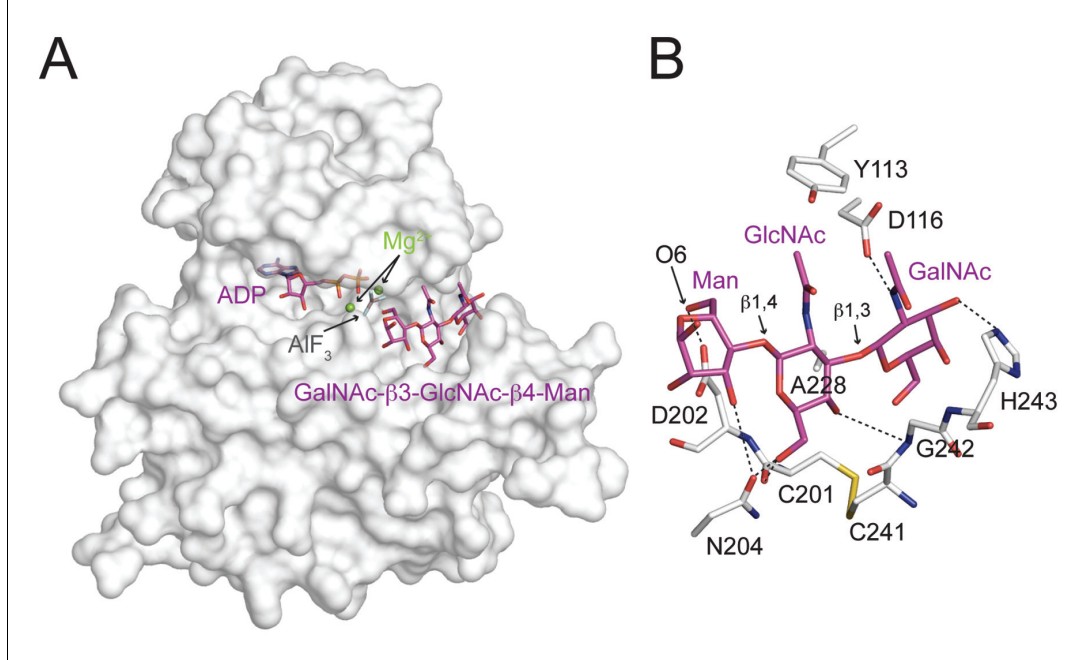

**Figure 6.** Interactions between DrPOMK and GalNAc-β3-GlcNAc-β4-Man. (A) Surface representation of DrPOMK in complex with $Mg^{2+}$ ions, ADP, $AlF_3$, and GalNAc-β3-GlcNAc-β4-Man. The same coloring scheme as in *Figure 3* is used. (B) An enlarged image of the GalNAc-β3-GlcNAc-β4-Man binding region showing the detailed molecular interactions important for the trisaccharide recognition. DrPOMK residues C201, D202, C241, G242, and H243 have both main chains and side chains shown as sticks, while the rest residues only have side chains depicted. Hydrogen bond interactions are shown as dashed lines. The linkages in the trisaccharide and the Man-O6 group are indicated.

The following figure supplement is available for figure 6:

**Figure supplement 1.** Electron density of Mg/ADP/$AlF_3$ and GGM-MU.

## Disease-causing mutations

Multiple mutations in POMK cause congenital or limb-girdle muscular dystrophy in humans (*Di Costanzo et al., 2014*; *Jae et al., 2013*; *von Renesse et al., 2014*). Besides the frameshift and nonsense mutations (F96fs, Q109*) that result in the loss of the majority of the kinase domain, three missense mutations have also been reported: L137R, Q258R, and V302D (*Figure 1A*). To understand how these alterations lead to disease, we modeled the structure of HsPOMK (*Figure 7*) based on that of DrPOMK using Swiss-Model (*Biasini et al., 2014*). Due to the high degree of sequence similarity between these two proteins, the resulting structural model shows a high GMQE (Global Model Quality Estimation) score (0.71), indicating its reliability. This gives us confidence to use this comparative model to gain mechanistic insights into the disease-causing mutants. Leu137 corresponds to Leu135[DrPOMK] and is involved in forming the regulatory spine structure (*Figure 2A*). Mutation of this residue to an Arg would disrupt this important internal hydrophobic network. Gln258 is located below the Cys203-Cys244 (Cys201[DrPOMK]-Cys241[DrPOMK]) disulfide bridge and points inside. Mutation to an Arg would collide with nearby residues including Lys284 and destabilize this region. Val302 is surrounded by hydrophobic residues including Trp175, Leu179, Leu293, and Leu338, which together anchor helix αH to the N- and C-terminal ends of helices αE and αF. Mutation to an Asp would severely disrupt the structure of the C-lobe. L137R failed to phosphorylate the GalNAc-β3-GlcNAc-β4-Man trisaccharide (*Yoshida-Moriguchi et al., 2013*). Q258R and V302D were also unable to phosphorylate the trisaccharide in vitro (*Figure 4A*), nor could they restore IIH6 immunoreactivity in vivo (*Figure 4B*), suggesting that these mutants are functionally defective. Finally, it is interesting to note that some patients with a POMK Q109* mutation presented with a limb-girdle muscle dystrophy (*Di Costanzo et al., 2014*), while others presented with a more severe congenital muscular dystrophy (*von Renesse et al., 2014*). This could suggest that there is a modifying gene for the

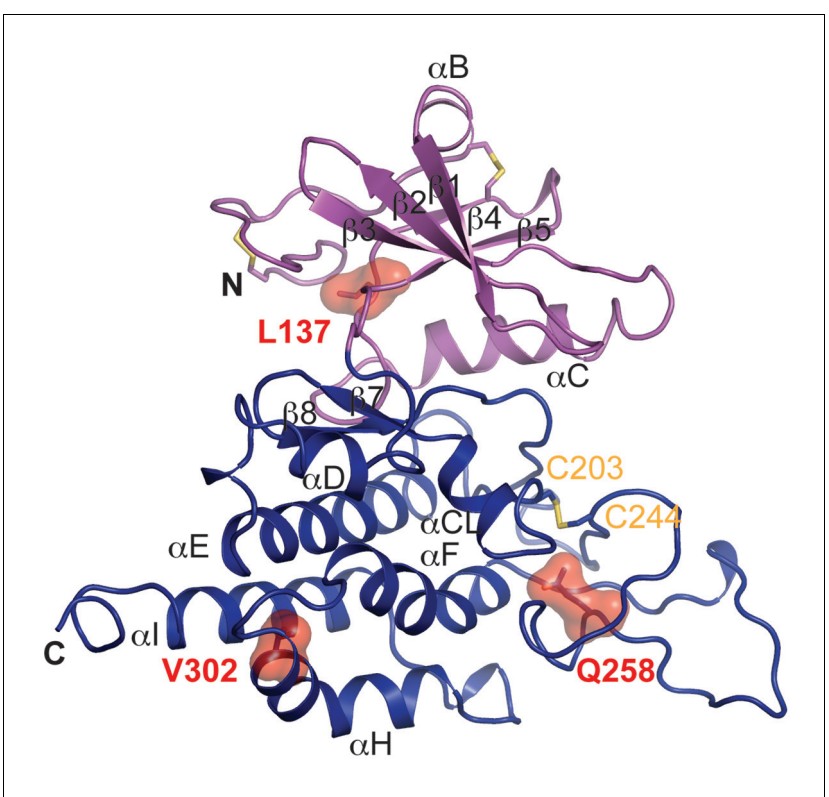

**Figure 7.** Structure modeling provides insights into POMK mutation related disease. Homology model of human POMK based on DrPOMK structure is shown as ribbon diagrams. The disease mutations are shown in surface representation and highlighted in red.

dystroglycanopathies, or a *POMK*-like gene that can compensate for the lack of *POMK*, or even there is exon skipping event that could result in a semi-functional POMK protein. Further studies are needed to precisely define the molecular pathology of this mutation.

## Discussion

POMK is unique among the 518 members of the human kinome in several aspects. First, it possesses bona fide kinase activity despite being annotated as a pseudokinase. Second, it contains a type II transmembrane domain and functions in the ER lumen. Third, it specifically phosphorylates an oligo-saccharide. Our results reported here reinforce the catalytic function of POMK and demonstrate that the active site of POMK is established by residues located in non-canonical positions, a phenomenon known as active site migration (*Kinch and Grishin, 2002*; *Todd et al., 2001*). POMK is not the first kinome member to display active site migration. The WNK (with no lysine [K]) kinases lack a Lys72[PKA]-equivalent residue in strand β3 and use a Lys in the Gly-rich loop similar to POMK (*Min et al., 2004*); and the atypical kinase haspin/Gsg2 (haploid germ cell-specific nuclear protein kinase/germ cell-specific gene-2) has a DY/FT motif to compensate for the absence of the DFG (*Eswaran et al., 2009*; *Villa et al., 2009*). POMK is unusual among the known kinases since none of the catalytically essential residues are found at conventional positions in the primary structure. Approximately 10% of the kinome members have been classified as pseudokinases, only a handful of which have been rigorously studied (*Eyers and Murphy, 2013*; *Manning et al., 2002*). The POMK study underscores the importance of combining sequence analyses with structural characterization to judge the potential activity and function of these proteins.

Most of the kinases residing in the cytosol and nucleus phosphorylate protein substrates. Our knowledge regarding glycan kinases remains in its infancy. Notably, among the thirteen secretory pathway kinases or kinase-like proteins discovered to date, two function as glycan kinases: Fam20B

and POMK. Fam20B specifically recognizes the Gal-β4-Xyl disaccharide and phosphorylates the xylose residue, whereas POMK phosphorylates the mannose in the GalNAc-β3-GlcNAc-β4-Man tri-saccharide. Both play critical roles to regulate glycan elongation: Fam20B for heparan sulfate and chondroitin sulfate proteoglycans, while POMK for α-DG. Phosphorylation of other types of extracellular glycans have been documented, whereas the responsible kinase remains elusive (*Breloy et al., 2012*). Identifying the substrates of uncharacterized secretory kinases and understanding their function could shed light on the elaborate glycobiology in the secretory pathway and human physiology related to glycans.

In summary, we have elucidated the molecular mechanism by which POMK catalyzes the phosphorylation of the trisaccharide GalNAc-β3-GlcNAc-β4-Man during the biosynthesis of functional α-DG. Our study provides mechanistic insights into a unique 'pseudokinase' and a deeper understanding of the molecular mechanisms that underlie the pathogenesis of muscular dystrophy caused by POMK mutations.

## Materials and methods

### Protein expression and purification

cDNA of HsPOMK and DrPOMK were purchased from Open Biosystems. cDNA of CoPOMK was prepared from the *C. owczarzaki* culture (a kind gift of Dr. Hiroshi Suga) using the Trizol reagent and TransScript Reverse Transcriptase (Transgen). For recombinant expression in insect cells, DNA fragments encoding HsPOMK (residues 49–350), DrPOMK (residues 49–347), and CoPOMK (residue 112–610) were cloned into the psMBP2 vector (*Tagliabracci et al., 2016*). Bacmids were generated using the Bac-to-Bac system (Invitrogen). Recombinant baculoviruses were generated and amplified using the sf21 insect cells (RRID: CVCL_0518), maintained in the SIM SF medium (Sino Biological Inc.). For protein production, Hi5 cells (RRID: CVCL_C190) grown in the SIM HF medium (Sino Biological Inc.) were infected at a density of 1.5–2.0×10$^6$ cells/ml. 48 hr post infection, 2 liters of conditioned medium were collected by centrifugation at 200 g. The medium was concentrated using a Hydrosart Ultrafilter (Sartorius) and exchanged into the binding buffer containing 25 mM Tris-HCl, pH 8.0, 200 mM NaCl. The proteins were then purified using the Ni-NTA resin (GE healthcare). Mutations were introduced into plasmids encoding HsPOMK by a PCR-based method, and the mutant proteins were purified similarly as the wild-type protein. For crystallization, TEV protease was used to remove the N-terminal 6xHis-MBP fusion tag from DrPOMK. Untagged DrPOMK was further purified by the anion exchange chromatography using a Resourse Q column (GE healthcare), followed by the size-exclusion chromatography using a Superdex 200 16/60 column (GE healthcare). To generate seleno-methionine (Se-Met) labeled DrPOMK, Hi5 cells were adapted to a methionine-free medium (Expression Systems) and infected with baculovirus. 100 mg/L Se-Met (Acros) was added to the medium at 12 and 36 hr post-infection. The Se-Met substituted protein was purified as described above.

### Production of GalNAc-β3-GlcNAc-β4-Man-α-MU (GGM-MU)

A large scale reaction was carried out using GlcNAc-β4-Man-α-MU (GM-MU) that had been produced by Sussex Research Labs (Canada) to make the final product GGM-MU. B3GALNT2dTM containing a 6xHis Tag was bound to metal affinity resin and added to 9 mM UDP-GalNAc (Sigma) and 9 mM GM-MU in 100 mM MES, pH 6.0, 10 mM MgCl$_2$, and 10 mM MnCl$_2$ and incubated for 48–72 hr at 37°C with rotation. At the end of this time there was about 70–80% conversion of substrate GM-MU to produce GGM-MU. The sample was then run over a C18 column (Supelcosil LC-18, 25 cm x 10 mm, 5 micron) with buffer A (50 mM ammonium formate, pH 4.0) and buffer B (80% Acetonitrile in buffer A). Using a 16% buffer B isocratic gradient and a flow rate of 3 ml/min, the product GGM-MU gave a signal at around 27 min by fluorescence (325 nm for excitation, and 380 nm for emission). This peak was collected, lyophilized, and brought up in ultra-pure water and quantitated using fluorescence and GlcA-MU (Sigma) as a standard.

### In vitro kinase assay

POMK kinase assay was performed as described in the Fam20B kinase assay (*Wen et al., 2014*; *Xiao et al., 2013*). Reactions were carried out in 50 mM HEPES, pH 7.5, 10 mM MnCl$_2$, 20 μM

GGM-MU, 100 μM [γ$^{32}$P]ATP (specific activity, 500 cpm/pmol) and 1 μg/ml POMK for 30 min at 20°C, and terminated with 20 mM EDTA and 15 mM ATP. The reaction mixtures were then loaded onto Sep-Pack C18 cartridges (Waters) pre-equilibrated with 0.2 M (NH$_4$)$_2$SO$_4$. Columns were washed with 2 ml of 0.2 M (NH$_4$)$_2$SO$_4$ for three times, and the substrates were eluted with 1 ml methanol. Incorporated radioactivity was measured by liquid scintillation counting (Tri-Carb 2810TR, PerkinElmer).

## Generation and characterization of HAP1 mutant cell lines

HAP1 cells (RRID: CVCL_Y019) are a haploid human cell line with an adherent, fibroblast-like morphology, originally derived from parent cell line KBM-7 (RRID: CVCL_A426). A protocol for generating HAP1 cells was previously published (*Carette et al., 2011*). HAP1 cells bearing a 10 bp deletion of exon 4 of the *protein O-mannose kinase* (*POMK*) gene, generated using the CRISPR/Cas9 system, were purchased from Horizon Discovery (HZGHC001338c004, clone 1338–4). The identity of the cells has been authenticated by the company using the STR profiling method. Mycoplasma testing of the cells were performed on a routine basis to ensure the cells are not contaminated. *POMK* knockout (KO) HAP1 cells lack the single copy of the wild-type *POMK* allele and are therefore null at the *POMK* locus. The sequence of the guide RNA used is TGAGACAGCTGAAGCGTGTT. Absence of the wild-type *POMK* allele was confirmed by Horizon Discovery, via PCR amplification and Sanger sequencing. PCR primers used for DNA sequencing are *POMK* Forward 5'-ACTTCTTCATCGCTCC TCGACAA-3', and *POMK* Backward 5'- GGATGCCACACTGCTTCCCTAA-3'.

## Adenovirus production

The open reading frame of human *POMK* was cloned into the *Bam*HI and *Xho*I sites of an expression plasmid with a C-terminal FLAG tag (pCCF) (*Tagliabracci et al., 2012*). *E1*-deficient recombinant adenoviruses (Ad5CMV-*POMK*-K93G, Ad5CMV-*POMK*-K93G/S108K, Ad5CMV-*POMK*-D227A, and Ad5CMV-*POMK*-D204A) were generated by the University of Iowa Viral Vector Core using the RAPAd system (*Anderson et al., 2000*). Assays for replication competence of adenoviruses were performed to check for contamination. Ad-*POMK*-WT, Ad-*POMK*-A230E, Ad-*POMK*-Q258R, and Ad-*POMK*-V302D were generated by ViraQuest Inc. (North Liberty, IA) using the RAPAd system (*Anderson et al., 2000*). Absence of the viral *E1* DNA sequence was confirmed by ViraQuest Inc. after PCR amplification of the viral DNA and staining on DNA agarose gel electrophoresis. Replication competence of adenoviruses was negative as assessed by plaque forming assays in cells performed from 10$^9$ viral particles up to 14 days.

## Cell culture and adenovirus infection

Wild-type C665 (a diploid cell line containing duplicated chromosomes of HAP1) and *POMK* KO HAP1 cells were maintained at 37°C and 5% CO$_2$ in Iscove's Modified Dulbecco's Medium (IMDM, Gibco) supplemented with 10% Fetal Bovine Serum (FBS) and 1% penicillin-streptomycin (Invitrogen). An average of 5.9×10$^6$ POMK KO HAP1 cells were infected in 2% IMDM on day 1, at 10 multiplicity of infection (MOI) of Ad5CMV-*POMK* wild-type or mutant, as indicated. On day 2, infection medium was replaced with 10% IMDM, and on day three the cells were processed for biochemical analyses.

## Glycoprotein enrichment and biochemical analysis

Cultured cells were washed twice in ice-cold Dulbecco's Phosphate-Buffered Saline (DPBS, Gibco), solubilized in 1% Triton X-100 in Tris-buffered saline (TBS, 50 mM Tris-HCl pH 7.6, 150 mM NaCl) with protease inhibitors (0.23 mM phenylmethylsulfonylfluoride and 0.64 mM benzamidine), and incubated in 200 μL wheat-germ agglutinin (WGA)-agarose (Vector Laboratories, AL-1023) as previously published (*Michele et al., 2002*). The following day samples were washed three times with 0.1% Triton X-100-TBS plus protease inhibitors, and heated to 99°C for 10 min with 250 μL of 5X Laemmli sample buffer. Samples were run on SDS-PAGE and transferred to PVDF-FL membranes (Millipore) as previously published (*Michele et al., 2002*). The mouse monoclonal antibody against the laminin-binding glycoepitope of α-DG (IIH6, Developmental Studies Hybridoma Bank, University of Iowa; RRID: AB_2617216) was characterized previously and used at 1:100 (*Ervasti and Campbell, 1991*). The rabbit polyclonal antibody, AF6868 (R and D systems; RRID: AB_10891298), was used at

a concentration of 1:200 for immunoblotting the core α-DG and β-DG proteins. Laminin overlay assays were performed as previously described (*Michele et al., 2002*). For immunoprecipitation of FLAG-tagged POMK, the flow-through from WGA pulldowns was incubated with 200 μL concanavalin A (Con A)-agarose (Vector Laboratories, AL-1003) slurry overnight at 4°C. The next day, samples were processed as for WGA pulldowns. A rabbit polyclonal anti-FLAG antibody (Sigma, F7425; RRID: AB_439687) was used at 1:500 to detect the FLAG epitope. Blots were developed with infrared (IR) dye-conjugated secondary antibodies (LI-COR Bioscience) and scanned using the Odyssey infrared imaging system (LI-COR Bioscience). Blot images were developed using the included Odyssey image-analysis software.

## Statistical analyses

The intensities of the protein bands on immunoblots were measured using the included Odyssey software and the raw integrated intensity values determined. The raw integrated intensity for IIH6 was divided by that of β-DG, and a IIH6: β -DG ratio was calculated for each sample. Within each replicate, the IIH6: β -DG ratio for each sample was normalized to the IIH6: β -DG ratio for the WT C665 sample. The means plus standard errors of the normalized IIH6: β -DG ratios from the three replicates were calculated using SigmaPlot 12.5. One-way ANOVA with the Dunnett's Method for Multiple Comparisons was performed, and the data for *POMK* KO sample set as the control. Differences were considered significant at a P-value less than 0.05. Graph images were created in Adobe Illustrator.

## Crystallization

DrPOMK in 25 mM Tris-HCl, pH 7.8, 200 mM NaCl was concentrated to 7 mg/ml and used for crystallization. The crystals were grown at 20°C using the hanging-drop vapor-diffusion method. Se-Met DrPOMK was crystallized in 0.5–0.6 M succinic acid, pH 7.0. The Se-Met crystals were transferred into a cryo-protection solution containing 40% (w/v) mannose, 0.6 M succinic acid, pH 7.0 and flash-frozen in liquid nitrogen. To obtain the Mg/ADP/AlF$_3$/GGM-MU complex crystal, the DrPOMK protein solution was supplemented with 20 mM MgCl$_2$, 10 mM ADP, 10 mM AlCl$_3$, and 40 mM NaF before mixed with the precipitant solution containing 0.3 M ammonium acetate, 0.1 M HEPES, pH 7.5, and 18% (w/v) PEG 3350. The crystals reached full size in 10–14 days, and were then transferred into a soaking solution containing 20 mM MgCl$_2$, 10 mM ADP, 10 mM AlCl$_3$, 40 mM NaF, 0.3 M ammonium acetate, 0.1 M HEPES, pH 7.5, 18% (w/v) PEG 3350, and 1 mM GGM-MU and soaked for 30 min. The crystals were then transferred into a cryo-protection solution (soaking solution plus 22% ethylene glycol) and flash-frozen in liquid nitrogen.

## Data collection and structure determination

All diffraction data were processed with HKL2000 (HKL Research). The structure of DrPOMK was determined by single-wavelength dispersion method using data collected from a Se-Met crystal. Heavy atom search, phase calculation and refinement, density modification, and initial model building were carried out with Phenix (*Adams et al., 2010*). The structural model was manually traced in Coot (*Emsley et al., 2010*). The transition state complex structure was determined by molecular replacement using the Se-Met structure in Phaser (*McCoy et al., 2007*) and refined using Phenix. Five percent randomly selected reflections were used for cross-validation (*Brünger et al., 1998*).

## Bioinformatics and structural analysis

Multiple sequence alignment of POMK homologues was performed using PROMALS3D (*Pei et al., 2008*). Structural alignment between POMK and PKA [PDB ID: 1 L3R (*Madhusudan et al., 2002*)] was performed using Dali (*Holm and Rosenström, 2010*). Interaction between GalNAc-β3-GlcNAc-β 4-Man and POMK was analyzed using PISA (*Krissinel and Henrick, 2007*). The structural model of human POMK was generated using Swiss-Model (*Biasini et al., 2014*). Molecular graphics were prepared using PyMol (Schrödinger, LLC).

## NMR spectroscopy

1D $^1$H NMR spectra of the trisaccharide GGM-MU in the absence and presence of DrPOMK were acquired at 25°C on a Bruker Avance II 800 MHz NMR spectrometer equipped with a cryoprobe

using a 50 ms $T_2$ filter consisting of a train of spin-lock pulses to eliminate the broad resonances from the protein (*Mayer and Meyer, 2001*). DrPOMK titrations were performed in 25 mM Tris (pH 8.0), 180 mM NaCl, and 10 mM MgCl$_2$ in 98% D$_2$O. The $^{13}$C and $^1$H resonances of the trisaccharide were reported previously (*Yoshida-Moriguchi et al., 2013*) and confirmed in the current study. The $^1$H chemical shifts are referenced to 2,2-dimethyl-2-silapentane-5-sulfonate. The NMR spectra were processed using NMRPipe (*Delaglio et al., 1995*) and analyzed using NMRView (*Johnson and Blevins, 1994*). The glycan binding affinity to DrPOMK was determined using glycan-observed NMR experiments as described recently (*Briggs et al., 2016*). For the resolved anomeric trisaccharide peak, the bound fraction was calculated by measuring the difference in the peak intensity in the absence (free form) and presence (bound form) of DrPOMK, and then dividing by the peak intensity of the free form. To obtain dissociation constant, the data were fitted to the standard quadratic equation using GraphPad Prism (GraphPad Software). The standard deviation from data fitting is reported.

## Acknowledgements

We are grateful to staff members of the Advanced Light Source (beamline 8.2.2) and Shanghai Synchrotron Radiation Facility (beamlines BL17U, BL19U) for assistance in X-ray data collection. We thank Hiroshi Suga of Hiroshima Prefectural University for the culture of *Capsaspora*, Shitang Huang of the Core Facilities at School of Life Sciences, Peking University for help with experiments involving [γ$^{32}$P]ATP, and Allison Bouska for assistance with DNA preparation. We thank Carolyn Worby, Alexandr Kornev, Susan Taylor, Erhard Hohenester, Jeffrey Esko, and members of the Xiao laboratory for insightful discussions and comments regarding the manuscript.

## Additional information

### Funding

| Funder | Grant reference number | Author |
|---|---|---|
| National Institute of Diabetes and Digestive and Kidney Diseases | DK18849 | Jack E Dixon |
| National Institute of Diabetes and Digestive and Kidney Diseases | DK18024 | Jack E Dixon |
| Howard Hughes Medical Institute | | Kevin P Campbell |
| Paul D. Wellstone Muscular Dystrophy Cooperative Research Center | 1U54NS053672 | Kevin P Campbell |
| National Natural Science Foundation of China | 31570735 | Junyu Xiao |
| National Key Research and Development Plan of China | 2016YFC0906000 | Junyu Xiao |

The funders had no role in study design, data collection and interpretation, or the decision to submit the work for publication.

### Author contributions

QZ, Carried out protein purification and crystallization, Collected diffraction data, Performed in vitro biochemical experiments; DV, Generated GalNAc-β3-GlcNAc-β4-Man-α-MU (GGM-MU); ASW, MEA, Generated recombinant adenoviruses, Performed in vivo functional studies of the POMK mutants; QF, NH, Performed molecular docking analyses on the interaction between GGM and DrPOMK; LNK, NVG, Performed bioinformatic and structural analyses; WW, XC, Attempted to chemically synthesize the GGM trisaccharide; LY, Measured the affinity of GGM-MU for DrPOMK using NMR spectroscopy and co-wrote the manuscript; JED, KPC, Co-designed the project, co-

wrote the manuscript, and supervised the research; JX, Performed crystallography and structural analyses, Co-designed the project, Co-wrote the manuscript

### Author ORCIDs

Kevin P Campbell, http://orcid.org/0000-0003-2066-5889

Junyu Xiao, http://orcid.org/0000-0003-1822-1701

## Additional files

### Major datasets

The following dataset was generated:

| Author(s) | Year | Dataset title | Dataset URL | Database, license, and accessibility information |
|---|---|---|---|---|
| Qinyu Zhu, David Venzke, Ameya S Walimbe, Mary E Anderson, Qiuyu Fu, Lisa N Kinch, Wei Wang, Xing Chen, Nick V Grishin, Niu Huang, Liping Yu, Jack E Dixon, Kevin P Campbell, Junyu Xiao | 2016 | Protein O-mannose kinase | http://www.rcsb.org/pdb/explore/explore.do?structureId=5GZA | Publicly available at the RCSB Protein Databank (accession no. 5GZA) |

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
