## [Decision Letter]

Thank you for submitting your article "Structure of protein O-mannose kinase reveals a unique active site architecture" for consideration by *eLife*. Your article has been favorably evaluated by Randy Schekman (Senior Editor) and three reviewers, one of whom, Reid Gilmore (Reviewer#1), is a member of our Board of Reviewing Editors. The following individuals involved in review of your submission have agreed to reveal their identity: Lance Wells (Reviewer #2); John D Scott (Reviewer #3).

The reviewers have discussed the reviews with one another and the Reviewing Editor has drafted this decision to help you prepare a revised submission.

Summary:

The manuscript from Zhu et al. describes the structure of protein O-mannose kinase POMK), an enzyme required for synthesis of the glycan on α-dystroglycan. The authors present very strong evidence that POMK is an ER-localized glycan kinase, rather than a pseudokinase. The previous designation of POMK as a pseudokinase was based upon the lack of any of the active site residues found in protein kinases at any of the expected positions. The crystal structure shows that the residues necessary for ATP binding and catalysis are located at other positions indicating that POMK displays active site migration. One interesting feature of the structure is that the active site is linked to the activation loop via a disulfide that is conserved in POMK. The authors have done a nice job of analyzing a panel of POMK mutants, both those based upon the crystal structure, and two patient alleles. Analysis of the structure based point mutants confirms the importance of residues that contact ADP-AlF3 and GalNAc-β3-GlcNAc-β4-Man. This manuscript also serves the role as presenting data for how a pseudokinase can in fact be an actual functional kinase and thus suggests that other pseudokinases should be investigated for kinase enzymatic activity. The paper is well written, the experiments follow a logical order, and the findings are of significant moment. There are only a few minor questions/comments that are listed below and enthusiasm for this manuscript is extremely high. The three reviewers indicated that this manuscript needed very minor modifications before it would be appropriate for *eLife*.

1) While the authors reference the Di Costanzo manuscript, they don't discuss the impact of the Q109* mutation – given that the individual with this mutation displayed LGMD instead of the expected WWS, this seems worth at least mentioning any potential rationale for patient phenotype.

2) Could the authors comment on why they did not use the NMR titration method to determine whether the POMK point mutants in the substrate binding site have a reduced affinity for the trisaccharide. Were these not conducted due to the NMR method needing large amounts of purified mutant protein?

3) A considerable strength of this article is a comparative mechanistic analysis that maps disease-causing mutations onto Danio Rerio POMK ortholog. These are important observations that provide new mechanistic insight into how this important glycan kinase works. I suggest that the authors more clearly emphasize that this is a comparative mechanistic model

*Reviewer #1:*

The manuscript from Zhu et al. describes the structure of protein O-mannose kinase POMK), an enzyme required for synthesis of the α-dystroglycan. The authors present very strong evidence that POMK is an ER-localized glycan kinase, rather than a pseudokinase. The previous designation of POMK as a pseudokinase was based upon the lack of any of the active site residues found in protein kinases at any of the expected positions. The crystal structure shows that the residues necessary for ATP binding and catalysis are located at other positions indicating that POMK displays active site migration. One interesting feature of the structure is that the active site is linked to the activation loop by a disulfide that is conserved in POMK. The authors have done a nice job of analyzing a panel of POMK mutants, both those based upon the crystal structure, and two patient alleles. Analysis of the structure based point mutants confirms the importance of residues that contact ADP-AlF3 and GalNAc-β3-GlcNAc-β4-Man. Overall, this is an interesting and important manuscript that includes very high quality experimental evidence. The manuscript has been written so that it will be accessible to a broad audience. I do not have any major concerns with the current manuscript.

*Reviewer #2:*

This manuscript by Zhu et al. focuses on the structural elucidation of POMK. Several years ago, it was determined that mutations in POMK were causal for multiple forms of CMD ranging from LGMD to WWS. It had also been previously determined that POMK catalyzed the transfer of a phosphate to a trisaccharide structure on α-dystroglycan that served as the core structure that becomes functionally glycosylated with matriglycan. However, what was extremely puzzling is that POMK fits in the family of pseudokinases that presumably did not have actual kinase activity. This manuscript addresses at the structural level this quandary. Here they clearly define the key residues involved in catalysis and make comparisons to PKA, a typical known well-defined protein kinase. In addition, they present data that explains some of the naturally occurring mutations seen in individuals with CMD. This manuscript also serves the role as presenting data for how a pseudokinase can in fact be an actual functional kinase and thus suggests that other pseudokinases should be investigated for kinase enzymatic activity. The paper is well written, the experiments follow a logical order, and the findings are of significant moment. There are only a few minor questions/comments that are listed below and enthusiasm for this manuscript is extremely high.

1) While the authors determine the Kd of the trisaccharide for PrPOMK, they do not use their eloquent NMR approach to determine the Kd for several of the mutants they generate that would likely impact substrate binding. Could these be generated to further confirm their mechanism of binding and the impact of key mutants? Were these not conducted due to the NMR method needing large amounts of purified mutant protein?

2) While the authors reference the Di Costanzo manuscript, they don't discuss the impact of the Q109* mutation – given that the individual with this mutation displayed LGMD instead of the expected WWS, this seems worth at least mentioning any potential rationale for patient phenotype.

*Reviewer #3:*

This is a well-written and interesting article that provides important new structural insight into the mechanism of action of Danio Rerio POMK an O-mannose kinase. The significance of this work comes from evidence that the human ortholog is defective in diseases of the basement membrane where defects arise in the biosynthesis of functional α-distroglycan. This is an important and well-controlled study that represents a valuable contribution to our understanding of the burgeoning family of enzymes. Of particular note, this study helps to re-designate as a glycan kinase rather than a pseudokinase. I only have a few comments.

It is slightly confusing to open the Results section with a diagram of the human POMK otholog followed by the crystal structure of the Danio Rerio POMK ortholog. This needs to clearly defined in the text. One solution would be to have a comparison of the human and D. Rerio othologs in Figure 1.

There is an element of redundancy in Figure 2. I wonder if it would be better to superimpose the Danio Rerio POMK ortholog over the backbone of the PKA structure. This might be a convenient way to depict the similarities and differences between the two structures.

A considerable strength of this article is a comparative mechanistic analysis that maps disease-causing mutations onto Danio Rerio POMK ortholog. These are important observations that provide new mechanistic insight into how this important glycan kinase works. I suggest that the authors more clearly emphasize that this is a comparative mechanistic model.

---

## [Author Response]

[…]

*1) While the authors reference the Di Costanzo manuscript, they don't discuss the impact of the Q109* mutation – given that the individual with this mutation displayed LGMD instead of the expected WWS, this seems worth at least mentioning any potential rationale for patient phenotype.*

We agree with the reviewer that one would expect a patient with a POMK Q109* mutation to have a severe WWS phenotype. However, it is not unusual that muscular dystrophy patients or animal models of muscular dystrophy to present with milder phenotypes. A possible reason for a milder phenotype could be expression of a modifier gene like was recently found in a mild dystrophic dog. Another reason could be the expression of a compensating gene like what is seen for utrophin. Finally, an exon skipping event in the disease gene can result in a smaller semi-functional protein being expressed like what is seen in revertant fibers that are positive for dystrophin.

In response, we have added the following sentences in the revised manuscript:

“Finally, it is interesting to note that some patients with a POMK Q109* mutation presented with a limb-girdle muscle dystrophy (Di Costanzo et al., 2014), while others presented with a more severe congenital muscular dystrophy (von Renesse et al., 2014). This could suggest that there is a modifying gene for the dystroglycanopathies, or a *POMK*-like gene that can compensate for the lack of *POMK*, or even there is exon skipping event that could result in a semi-functional POMK protein. Further studies are needed to precisely define the molecular pathology of this mutation.”

*2) Could the authors comment on why they did not use the NMR titration method to determine whether the POMK point mutants in the substrate binding site have a reduced affinity for the trisaccharide. Were these not conducted due to the NMR method needing large amounts of purified mutant protein?*

The NMR method indeed needs a lot of purified protein, which is difficult to obtain for POMK, since it requires an eukaryotic expression system. Also, the weaker the binding of the mutants, the more proteins are needed. The three mutants we generated in the substrate binding site based on the structural analyses all have greatly impaired or abolished kinase activity in vitro(D118A, A230E, C203A/C244A, Figure 4), so we expect their binding to the trisaccharide will be significantly decreased. Thus, it will be difficult to perform the NMR titration experiment on these mutants.

*3) A considerable strength of this article is a comparative mechanistic analysis that maps disease-causing mutations onto Danio Rerio POMK ortholog. These are important observations that provide new mechanistic insight into how this important glycan kinase works. I suggest that the authors more clearly emphasize that this is a comparative mechanistic model*

We thank the reviewer for this suggestion. In the revised manuscript, we have more clearly emphasized that our discussion on the disease-causing mutants is based on a structural model of human POMK and included more details regarding the model quality:

“To understand how these alterations lead to disease, we modeled the structure of HsPOMK (Figure 7) based on that of DrPOMK using Swiss-Model (Biasini et al., 2014). Due to the high degree of sequence similarity between these two proteins, the resulting structural model shows a high GMQE (Global Model Quality Estimation) score (0.71), indicating its reliability. This gives us confidence to use this comparative model to gain mechanistic insights into the disease-causing mutants.”

*Reviewer #3:*

*This is a well-written and interesting article that provides important new structural insight into the mechanism of action of Danio Rerio POMK an O-mannose kinase. The significance of this work comes from evidence that the human ortholog is defective in diseases of the basement membrane where defects arise in the biosynthesis of functional α-distroglycan. This is an important and well-controlled study that represents a valuable contribution to our understanding of the burgeoning family of enzymes. Of particular note, this study helps to re-designate as a glycan kinase rather than a pseudokinase. I only have a few comments.*

*It is slightly confusing to open the Results section with a diagram of the human POMK otholog followed by the crystal structure of the Danio Rerio POMK ortholog. This needs to clearly defined in the text. One solution would be to have a comparison of the human and D. Rerio othologs in Figure 1.*

We thank the reviewer for this helpful suggestion. We made a new Figure 1 that includes schematic representations of both human POMK and *Danio Rerio* POMK.

*There is an element of redundancy in Figure 2. I wonder if it would be better to superimpose the Danio Rerio POMK ortholog over the backbone of the PKA structure. This might be a convenient way to depict the similarities and differences between the two structures.*

We made a new figure with POMK and PKA superimposed. Indeed, this new figure better depicts the structural similarities and differences between these two proteins. On the other hand, it is difficult to incorporate all the details regarding these two structures in this new figure (labels for the secondary structures, regulatory spines, etc.). In the revised manuscript, we have kept the original Figure 2 and included this new figure as Figure 2—figure supplement 1.